# High-order species interactions shape ecosystem diversity

Eyal Bairey[1], Eric D. Kelsic[2,†] & Roy Kishony[2,3]

Classical theory shows that large communities are destabilized by random interactions among species pairs, creating an upper bound on ecosystem diversity. However, species interactions often occur in high-order combinations, whereby the interaction between two species is modulated by one or more other species. Here, by simulating the dynamics of communities with random interactions, we find that the classical relationship between diversity and stability is inverted for high-order interactions. More specifically, while a community becomes more sensitive to pairwise interactions as its number of species increases, its sensitivity to three-way interactions remains unchanged, and its sensitivity to four-way interactions actually decreases. Therefore, while pairwise interactions lead to sensitivity to the addition of species, four-way interactions lead to sensitivity to species removal, and their combination creates both a lower and an upper bound on the number of species. These findings highlight the importance of high-order species interactions in determining the diversity of natural ecosystems.

[1] Department of Physics, Technion—Israel Institute of Technology, Haifa 3200003, Israel. [2] Department of Systems Biology, Harvard Medical School, Boston, Massachusetts 02115, USA. [3] Department of Biology and Department of Computer Science, Technion—Israel Institute of Technology, Haifa 3200003, Israel. † Present address: Department of Genetics, Harvard Medical School, Boston, Massachusetts 02115, USA. Correspondence and requests for materials should be addressed to R.K. (email: rkishony@technion.ac.il).

A major challenge of theoretical ecology is explaining the stable coexistence of multi-species communities[1–5]. While species interactions are important for maintaining natural diversity[1,6–12], their random nature destabilizes large communities. Even a community that satisfies the exclusion principle[13], or is otherwise inherently stabilized[6], can have its stability jeopardized by random interactions among species[11]. This challenge was embodied in the seminal work by May[14], showing that the number of species that can stably coexist in randomly interacting communities is inversely proportional to the strength of their effective pairwise interactions[15]. In ecological communities with strictly random pairwise interactions, this result predicts an upper threshold on diversity, as observed in random pairwise interaction models[16,17]. Constrained properties of interactions, such as trophic level[18,19], intervality, broad degree distributions[20], positive and negative reciprocity[21–23], distribution skewness[24,25] and connectance between differentially self-regulating species[26] have been suggested as ways to explain the coexistence of large communities. Less is known about the feasibility, and possibly even the necessity, of diversity within the strict context of unconstrained random interactions.

Species can exhibit interactions that inherently involve multiple species[27]. While modelling in ecology often assumes species communities with pairwise interactions (Fig. 1a), species can also interact in higher-order combinations, that is, the interactions between two species can be modulated by other species[27–30]. For instance, while a microbial species can produce an antibiotic that inhibits the growth of a competing species, this pairwise inhibitory effect can be attenuated by a third species that produces an enzyme that degrades the antibiotic[31–33]. This third species thus modifies the interaction between the antibiotic-producing and sensitive species, without having a direct effect on any of them in isolation (Fig. 1b). The expression or activity of the antibiotic-degrading enzyme may in turn be inhibited by compounds produced by a fourth species[34], thereby generating a four-way interaction (Fig. 1c). Another general class of high-order interactions may arise when species exhibit adaptive behaviour[30,35], such as a predator switching its prey when a more preferred prey becomes available[36], or a prey that reacts to the presence of a predator by decreasing its foraging activity directed at a third species[37]. Such behavioural effects can quickly lead to interactions at even higher order; for example, the third species may subsequently increase its activity against a

fourth species, generating an effect that is inherently four-way[28,38]. While the paradigm of pairwise interactions can capture the density-mediated effect of a predator on a species devoured by its prey that arises from the predator's impact on the prey's density, it misses the often larger trait-mediated effect[28,37,39] of the predator on this third species through its impact on the prey's behaviour.

While the importance of high-order interactions has been recognized, their general impact on critical community size has not been studied. One class of three-way interactions, whereby one species attenuates the negative interactions between two others, can stabilize well-mixed communities in specific network configurations[31]. Three-way interactions have further been shown to promote diversity in patchy environments by increasing sensitivity to initial conditions[40]. While increasing the order of interactions weakens their destabilizing effect[41] and can increase the variance of species abundances at stationary states[42], it is unclear how high-order interactions affect the critical threshold on community size[28,43], and whether the classical result that random interactions impose a cap on community diversity still applies.

Here, we combine simulations and theoretical analysis of random communities to examine the relation between diversity and stability in ecosystems with both pairwise and higher-order interactions. We find that high-order interactions impose a lower bound on the number of species, in addition to the upper bound imposed by pairwise interactions. Altogether, interactions of different orders therefore define an optimally stable range of diversities.

## Results

**Modelling communities with pairwise and high-order interactions.** We use a replicator dynamics model[16,23,44,45] to simulate communities with interactions of different orders[41,42,46]. Our model assumes $N$ species with abundances $x_i$, and whose per-capita growth rates $\dot{x}_i/x_i$ are determined by the sum of pairwise interactions $A_{ij}x_j$ (where $A_{ij}$ determines the effect of species $j$ on species $i$), of three-way interactions $B_{ijk}x_jx_k$ (where species $j$ and $k$ have a joint effect on species $i$) and so on. This model therefore represents the Taylor expansion of a general system that includes trait-mediated effects as well as pairwise interactions:

$$\frac{\dot{x}_i}{x_i} = f_i(\vec{x}) - \sum x_j f_j(\vec{x}),$$

$$f_i(\vec{x}) = -x_i + \sum_{j=1}^{N} A_{ij}x_j + \sum_{j=1}^{N}\sum_{k=1}^{N} B_{ijk}x_jx_k + \quad (1)$$

$$\sum_{j=1}^{N}\sum_{k=1}^{N}\sum_{l=1}^{N} C_{ijkl}x_jx_kx_l + \ \ldots$$

The model assumes random interactions perturbing an inherently stable community. The stability is provided by the first term ($-x_i$), which reflects the assumption that species are self-limiting in high concentrations. The pairwise interactions matrix is given by $A = \sqrt{\alpha}\tilde{A}$, where $\tilde{A}$ is a random matrix whose elements are drawn from a Gaussian distribution with mean 0 and variance 1, and $\alpha$ determines the strength (variance) of the pairwise interactions. The three-way interactions are set by the tensor $B = \sqrt{\beta}\tilde{B}$, where $\tilde{B}$ is a random three-dimensional ($N \times N \times N$) tensor whose elements are drawn from a Gaussian distribution with mean 0 and variance 1, and $\beta$ represents the strength of three-way interactions; and similarly with four-way interactions whose strength scales with a parameter $\gamma$. As the total biomass is often determined by external resources, such as nutrients, energy and space, we keep the sum of all species abundances constant by subtracting from the individual growth rate of each species the average growth rate of the species in the community.

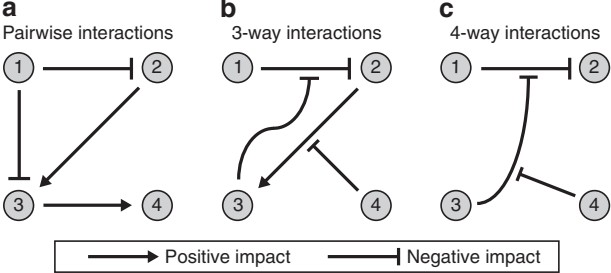

**a** Pairwise interactions
**b** 3-way interactions
**c** 4-way interactions

→ Positive impact    ⊣ Negative impact

**Figure 1 | Species can exhibit high-order interactions, whereby the interaction between two species is affected by other species.** (**a**) Pairwise interactions in a community: species affect each other directly. For instance: species '1' could be producing an antibiotic, inhibiting the growth of species '2'. (**b**) Three-way interactions in a community: one species can modulate the interactions between two others. For instance, species '3' might degrade the antibiotics produced by species '1', thereby attenuating the inhibitory effect of species '1' on species '2'. (**c**) Four-way interactions in a community. For example, species '4' might produce a compound that inhibits the antibiotic-degrading enzyme produced by species '3'.

**High-order interactions set a lower bound on diversity.** As expected from the analysis by May[14] and subsequent results[16,17], without high-order interactions we find that coexistence is lost when the pairwise interaction strength α is increased beyond a critical threshold that decreases with the number of species. We ran simulations with strictly pairwise interactions (α > 0 with β = 0, γ = 0) for varying interaction strengths α and different numbers of species N. When the pairwise interaction strength α is small the simulations reach a steady state where all species coexist (Supplementary Fig. 1a), but when α increases simulations start exhibiting species extinctions (Supplementary Fig. 1b). Therefore, for each number of species N, we can define a critical threshold $\alpha_c$ as the strength of pairwise interactions at which community feasibility (positive abundance for all species at equilibrium[24]) is lost for a given fraction of the simulations (5%, Supplementary Fig. 1c,d). When the number of species N is increased, the critical threshold $\alpha_c$ decreases as $1/N$, in accordance with the result by May[14] (Fig. 2a). Thus, as communities become more diverse they are more sensitive to random pairwise interactions (Fig. 2b).

Strikingly, this classic relation between diversity and stability is inverted when high-order interactions are considered. We repeated the analysis above with strictly three-way interactions (only β > 0) and strictly four-way interactions (only γ > 0). Similar to the case of pairwise interactions, when the interaction strengths β or γ are increased beyond a certain level, $\beta_c$ or $\gamma_c$, feasibility is lost and extinctions abound. However, in contrast to pairwise interactions, the critical thresholds for high-order interactions do not decrease with the number of species. The critical threshold of three-way interactions $\beta_c$ remains unchanged, while the critical threshold of four-way interactions $\gamma_c$ increases in proportion to the number of species (Fig. 2a). These four-way interaction communities can therefore withstand stronger interactions the more species they have. For a fixed strength of four-way interactions, these communities are stabilized, rather than destabilized, as the number species increases (Fig. 2c; increased stability with diversity can also appear in reciprocity models[21]). Thus, while pairwise interactions put a higher bound on diversity, it is a lower bound that is created by high-order interactions. Provided that the total abundance is constant (does not change with the number of species), these results are robust to various changes in the underlying model: they hold for different distributions of the random coefficients (Supplementary Fig. 2), for connectance values smaller than 1 (Supplementary Fig. 3) and when high-order diagonal terms are set to zero ($C_{ijk} \neq 0$ only when $i \neq j \neq k \neq i$; Supplementary Fig. 4). Similar scalings of the critical strengths of the different interaction orders with the number of species also occur when the dynamic equations are replaced by a Lotka–Volterra model (Supplementary Fig. 5).

**Diversity scales differently with different interaction orders.** To better understand the relation between the diversity of a community and its sensitivity to interactions of different orders, we examine a modified pairwise interactions matrix that captures the high-order interactions around a fixed point. Considering an approximation for how the interactions of different orders contribute to the effective pairwise interactions considered in the analysis by May[14] (namely, the Jacobian), we derive a scaling relationship for the dependence of the feasible range of N on the different orders of interactions (see the 'Methods' section, Supplementary Fig. 6):

$$\alpha + \frac{\beta}{N} + \frac{\gamma}{N^2} + \ldots < \frac{1}{N}. \qquad (2)$$

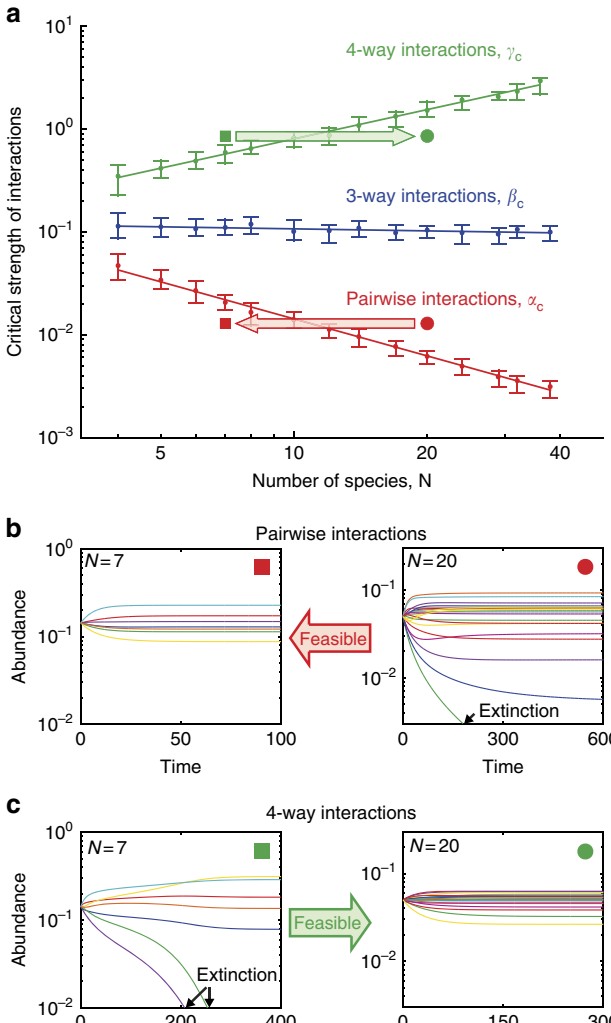

**Figure 2 | The destabilizing effect of diversity is inverted when interactions are high-order.** (**a**) The critical strength of interactions beyond which the community becomes unfeasible (defined as the value at which 5% of random communities exhibit extinctions; error-bars indicate the range of 2–10%). While the critical strength of pairwise interactions decreases with community diversity ($\alpha_c$, red, slope = −1.20 ± 0.05), it remains unchanged for three-way interactions ($\beta_c$, blue, slope = −0.07 ± 0.04) and increases for four-way interactions ($\gamma_c$, green; slope = 0.95 ± 0.04). (**b**) Example simulations of two communities with pairwise interactions of the same strength but with different numbers of species (N = 7, left; N = 20, right; red square and circle in **a**). While the small community shows convergence to a stable fixed-point with all species coexisting, the large community exhibits species extinction. (**c**) As shown in **b**, but for communities with four-way interactions. Here the trend is reversed, and the small community exhibits extinctions, while the large community exhibits stable coexistence.

Intuitively, this condition tests whether the s.d. of the sum of all random interactions affecting a species near a fixed point is smaller than the self-stabilizing term (assuming fixed total abundance, see the 'Methods' section).

This theoretical analysis explains the different scaling of each of the interaction orders with N in accordance with our simulation results (Fig. 2a). When high-order interactions are turned off (only α > 0), the critical strength of pairwise interactions $\alpha_c$ decreases as $1/N$. The critical threshold of exclusively three-way

interactions $\beta_c$ does not depend on the number of species, while that of strictly four-way interactions $\gamma_c$ is proportional to the number of species. Thus, interactions of any given order destabilize communities, and the effects of different orders are additive, but they scale differently with diversity. Small differences between these theoretical slopes and the numerical slopes of Fig. 2a might result from the small species numbers of the simulations, for which the analytic analysis holds only approximately.

**Mixed interactions define range of allowed diversities.** Since communities with a small number of species are sensitive to high-order interactions, while communities with a large number of species become sensitive to pairwise interactions, we asked whether the combination of pairwise and high-order interactions might dictate an intermediate range of species numbers that optimizes feasibility. To answer this question, we ran simulations with both pairwise and four-way interactions for three different community sizes $N$ (for simplicity we set $\beta = 0$; the effect of $\beta > 0$ is discussed below). For each value of $N$, we scanned $\alpha$ and $\gamma$ and identified the domain where communities are feasible (Fig. 3a). Since the critical value of $\alpha$ decreases with $N$, while the critical value of $\gamma$ increases, there are regions where an intermediate-sized community would be feasible, while either smaller or larger communities would not (area within the $N = 8$ domain but not within the $N = 5$ or the $N = 18$ domain). To understand how the feasibility of specific communities depends on the number of species, we ran simulations where we gradually add species with random interactions to communities with given values of interaction strengths $\alpha$ and $\gamma$: pairwise-dominated,

four-way dominated and mixed. When pairwise interactions are dominant, feasibility decreases when $N$ is increased beyond a threshold (Fig. 3b). In contrast, when four-way interactions are dominant, the community becomes sensitive to the removal of species (Fig. 3d). In the mixed case, when pairwise and high-order interactions are combined, feasibility is maximized for an intermediate number of species; both the removal and the addition of species destabilize the community (Fig. 3c).

The combination of pairwise and high-order interactions thus defines a range of stable diversities. Assuming that a given natural ecosystem can be characterized by its average strengths of interactions of different orders ($\alpha$, $\beta$, $\gamma$...), which are characteristics of its species and the environment, the upper and lower bounds on its diversity $N$ can be derived as the roots of equation 2 (see the 'Methods' section; Supplementary Fig. 7). As the total strength of interactions $\alpha + \beta + \gamma$ increases, these upper and lower bounds grow closer and the range of allowed diversities narrows around a defined number of species $N^*$ (Fig. 4 for $\beta = 0$; using $\beta = \gamma$ gives qualitatively similar results, Supplementary Fig. 8). This value of optimally stable diversity is determined by the relative strengths of the four-way and pairwise interactions ($N^* = \sqrt{\gamma/\alpha}$, see the 'Methods' section). High-order dominated and strong interactions can thus require communities to maintain a large number of species to remain feasible.

## Discussion
The combination of pairwise and high-order interaction strengths determines both upper and lower bounds on the diversity of multi-species communities. While very little is known

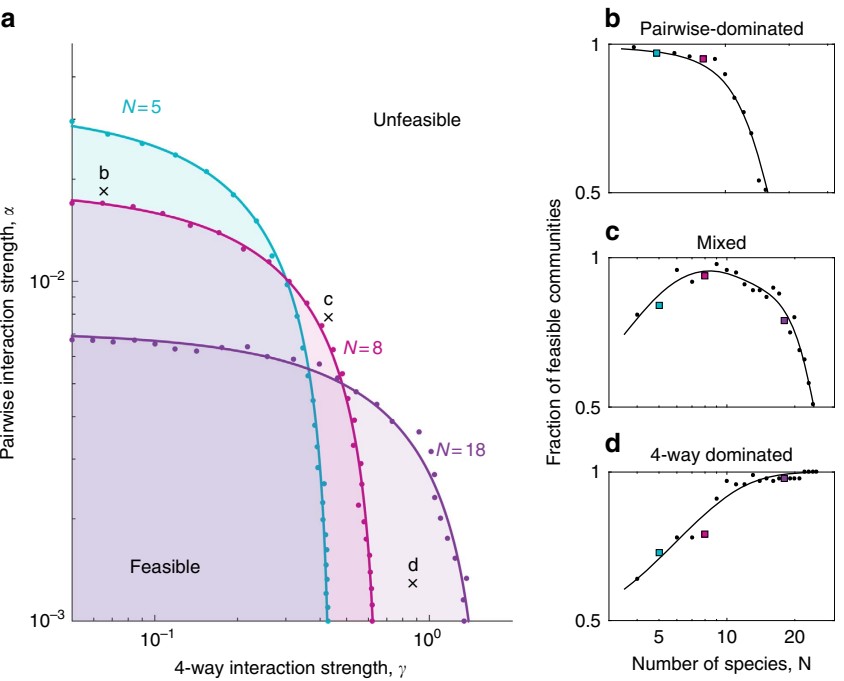

**Figure 3 | The combination of high-order and pairwise interactions determines an optimally stable range of diversities.** (**a**) Regions of stability for 5, 8 and 18 species communities in the space of pairwise $\alpha$- and four-way $\gamma$-interaction strengths (assuming no three-way interactions, $\beta = 0$). Dots show the critical thresholds at which 5% of the simulations become unstable when the interaction strengths increase along radial lines in the log$\alpha$–log$\gamma$ space (see the 'Methods' section). (**b–d**) The fraction of stable communities over the initial number of species is shown for three combinations of pairwise and four-way interaction strengths (at the points labelled by 'x' in panel **a**); pairwise-dominated in **b**, mixed pairwise and four-way in **c**, and four-way dominated in **d**. When pairwise interactions dominate, diversity is bounded from above; when four-way interactions dominate, a lower bound on diversity appears; with both types of interactions, the level of diversity is defined to a narrow range, and within this range the community is sensitive not only to the addition but also to the removal of species.

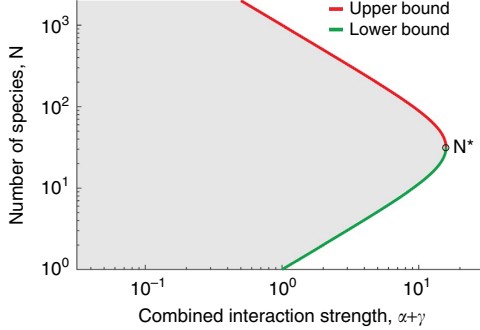

**Figure 4 | Range of stable diversities over interaction strength for a community with pairwise and four-way interactions.** As the total interaction strength is increased, the lower bound on the number of species increases, while the upper bound decreases, until the range of allowed diversities narrows around a defined number of species $N^*$. Here, the relative strengths of pairwise and four-way interactions is fixed so that both will have a significant effect ($\gamma = 10^3\alpha$). A similar behaviour appears when accounting for three-way interactions (Supplementary Fig. 8). Note that an ecologically relevant lower bound ($N \gg 1$) requires the coefficient of the high-order interactions to be much larger than the pairwise interactions ($\gamma \gg \alpha$).

about the abundance and strength of high-order interactions in nature, these results bring a new perspective to understand much empirical evidence showing that communities are sensitive not only to the addition of species but also to the opposite perturbation, namely the removal of species[47–50]. Considering the perplexing effects of high-order interactions may allow a better understanting of why high diversity is often a necessity rather than an option for natural ecosystems.

## Methods

**Numerical simulations of community dynamics.** To find the probability that a random community made up of $N$ species with specified strengths of pairwise and high-order interactions is feasible, we numerically integrate equation 1 with different randomizations of the interaction matrices. For a given $N$, we generate a set of random pairwise, three- and four-way interaction matrices $\{\tilde{A}, \tilde{B}, \tilde{C}\}$, and then for a given choice of $\alpha, \beta, \gamma$ we substitute into equation 1: $A = \sqrt{\alpha}\tilde{A}$, $B = \sqrt{\beta}\tilde{B}$ and $C = \sqrt{\gamma}\tilde{C}$. We then follow the species abundances as they change from a uniform initial value ($x_i = 1/N$; in the stable regime, the equations should not be sensitive to initial conditions[45]) using the MATLAB ode45 solver with default integration properties. Simulations were ended when they reach a persistent state (steady-state or bounded oscillations). We repeat this process $R$ times ($\sim 300$) with different sets of random pairwise, three- and four-way interaction matrices.

We concluded that a community had reached a persistent state by demanding that the entropy ($-\Sigma_i x_i \log(x_i)$) remain constant or bounded in its fluctuations. We evolve the equations for a fixed number of time units ($\Delta t = 10$), and after each of those intervals, we calculate the ratio between the minimal value taken by the species distribution entropy in two different periods of time: the last tenth and the last three-tenths of the simulation time length. If the ratio between those two values, as well as the similar ratio between the entropy maxima in those two periods, falls within a range of $\epsilon = 10^{-5}$ of 1, we concluded the simulation. If this condition was not met after a long time ($10^4$ time units) we concluded the simulation as well, and to account for the possibility that a species would have become extinct after this time threshold, we counted those simulations as unfeasible in the calculation of the lower error-bars.

We defined a community as feasible if all the species existed at the end of the simulation, with an abundance above a defined threshold (set as $10^{-5}/N$). The probability that a community would be feasible was then defined as the fraction of stable communities out of the $R$ random communities simulated. To find the critical strength of interactions for a given number of species, we used a fixed collection of $R$ sets of pairwise, three- and four-way interaction matrices $\{\tilde{A}, \tilde{B}, \tilde{C}\}$ and increased the interaction parameters until 5% of the communities exhibited extinctions. In Fig. 2 (as well as Supplementary Figs 2–5) this was done by increasing the strength of interactions of a given order, while the rest are kept at 0. In Fig. 3a this was done along radial lines in log ($\alpha$), log ($\gamma$) space:

$\alpha = 0.001 \cdot 1.2^{r \cdot \cos\theta}$, $\gamma = 0.05 \cdot 1.3^{r \cdot \sin\theta}$, such that for each value of $\theta$, we increased $r$ until stability was lost.

**Deriving stability criterion by considering effective pairwise interactions.** The Jacobian of the system at a given point is:

$$J_{ij} = \delta_{ij}\left(f_i - \sum x_j f_j\right) + x_i\left(-\delta_{ij} + A_{ij} + \sum_k x_k\left(B_{ijk} + B_{ikj}\right)\right.$$
$$+ \sum_{k,l} x_k x_l\left(C_{ijkl} + C_{ikjl} + C_{iklj}\right) + \cdots$$
$$+ 2x_j - \sum_k x_k\left(A_{jk} + A_{kj}\right) - \sum_{k,l} x_k x_l\left(B_{jkl} + B_{kjl} + B_{klj}\right) + \cdots\Big)$$

Where, $\delta_{ij} = \begin{cases} 1 & i = j \\ 0 & i \neq j \end{cases}$ is the Kronecker delta. At a fixed point, the first term vanishes. We may also neglect the terms in the last line which are derivatives of the normalization, as we will justify below. Assuming stability of all species coexisting, the value of species abundances at this fixed point must scale as $x_i \approx 1/N$ (because the total abundance is fixed to 1). Equation 1 can therefore be approximated as pairwise dynamics:

$$\dot{x}_i = x_i \sum_{j=1}^{N} A_{ij}^{\text{eff}} x_j$$

with

$$A_{ij}^{eff} \approx -\delta_{ij} + \sqrt{\alpha}\tilde{A}_{ij} + \sqrt{\beta}\sum_{k=1}^{N}\frac{\tilde{B}_{ijk} + \tilde{B}_{ikj}}{N} + \sqrt{\gamma}\sum_{k=1}^{N}\sum_{l=1}^{N}\frac{\tilde{C}_{ijkl} + \tilde{C}_{ikjl} + \tilde{C}_{iklj}}{N^2} + \cdots$$

the fixed point will be stable if the eigenvalues of the Jacobian all have negative real parts. Within our assumption $x_i \approx 1/N$, we may focus on the eigenvalues of $A^{\text{eff}}$, which differs from the Jacobian by a constant factor $N$ that does not affect the signs of the eigenvalues (as previously reported[15], the actual Jacobian in a steady-state deviates somewhat from this simple analysis due to deviation of the fixed-point $x_i$ from $1/N$; Supplementary Fig. 9).

The entries of the random component of $A^{\text{eff}}$ are sums of independent identically distributed Gaussians with 0 mean, and the variance of such a sum is additive. For simplicity, we assume that only elements with $j > k > l$ are non-zero (such as in Supplementary Fig. 4), so the three-way interactions term averages over $N-1$ such elements and its variance is therefore $\sim N$ times smaller, and similarly with the higher-order terms. The variance of the random component of $A^{\text{eff}}$ is thus given by:

$$\alpha + \frac{\beta}{N} + \frac{\gamma}{N^2} + \cdots$$

by Girko's circular law, as $N \to \infty$, the eigenvalues of a random matrix M of order $N \times N$ will form a disk of radius $\sqrt{\text{var}(M) \cdot N}$ around the origin on the complex plane. In a large enough community, and taking into account the stabilizing term, the eigenvalues of $A^{\text{eff}}$ will therefore be approximately distributed in a disc around $(-1, 0)$ (Supplementary Fig. 6), so that the community will be stable with high probability if the radius of the disc is smaller than 1, or:

$$\alpha + \frac{\beta}{N} + \frac{\gamma}{N^2} + \cdots < \frac{1}{N}$$

If we remove the assumption that only elements with $j > k > l$ are non-zero, the high-order interaction strengths in the last equation are multiplied by constant factors, but the scaling is unaffected.

This analysis does not depend on the specific distribution of the matrix elements[51], as long as their variance is fixed, and our simulations indeed show it holds for different distributions (Supplementary Fig. 2). Similar scaling results can be obtained relying on Gershgorin's circle theorem (see ref. 52 for the case of pairwise interactions; generalization to high-order case follows from the same reasoning applied above). The normalization term should not have a significant impact on the instantaneous stability analysis, since by the same reasoning, its contribution to the variance of the Jacobian would be $N$ times smaller than the interaction term, as it is itself an average of $N$ interaction terms. However, this analysis does depend on the sum of species abundances being constant (and $x_i$ thereby scaling as $1/N$); a different scaling of critical interaction strengths will appear when the total abundance scales with the number of species[41,42,46].

Note that local stability analysis models exclusively pairwise interactions, since the equation is linearized around a point and the high-order interactions are therefore embedded in the coefficients of the effective pairwise interactions obtained in the linearization. Our analysis therefore extends the result by May[14] in the sense that it suggests how the interaction strength in the Jacobian May analyses should scale with the number of species with respect to the strengths of interactions of different orders.

**Deriving stability criterion by analysis of interaction variance.** The same scaling of diversity with interactions of different orders can also be obtained by

demanding that the expected variance of the random interactions be weaker than the self-stabilizing term. The $n$-species interaction term is given by:

$$\sqrt{\alpha^{(n)}} \sum_{i_1, i_2, \ldots, i_{n-1}}^{N} \widetilde{A^{(n)}} x_{i_1} x_{i_2} \cdots x_{i_{n-1}}$$

with $\widetilde{A^{(n)}}$ an $n$-dimensional interaction tensor whose variance is 1, and $\alpha^{(n)}$ the strength of the $n$-species interactions. Since the value of species abundances near a fixed point with all species coexisting scales as $x_i \approx 1/N$, and from the additivity of the variance of independent random variables, the standard deviation of such a term scales as $\sqrt{\alpha^{(n)} N^{-(n-1)}}$. Comparing this value with the stabilizing term $|-x_i| \approx 1/N$, we get the condition $\alpha^{(n)} < N^{n-3}$, in agreement with our result ($\alpha < 1/N$ for pairwise interactions, $\beta < 1$ for three-species interactions and so forth). The additivity of the variance allows us to extend this reasoning to obtain equation 2 when interactions of different orders are combined.

**The optimal community size $N^\star$.** Restricting interactions up to the fourth order, communities are stable if they satisfy the equation:

$$\alpha + \frac{\beta}{N} + \frac{\gamma}{N^2} < \frac{1}{N}$$

demanding an equality and solving for $N$ gives the lower and upper bounds for number of species:

$$N_{\min,\max} = \frac{1 - \beta \pm \sqrt{(1-\beta)^2 - 4\alpha\gamma}}{2\alpha}$$

these two bounds narrow around a defined optimal number of species when the discriminant vanishes: $1 - \beta = 2\sqrt{\alpha\gamma}$, which gives $N^* = \sqrt{\gamma/\alpha}$.

**Code availability.** MATLAB code for the replicator and Lotka–Volterra models is available from the corresponding author on request.

**Data availability.** The simulation results are available from the corresponding author on request.

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

## Acknowledgements

We thank Ariel Amir, Michael Baym, Guy Bunin, Michael Elowitz, Renan Gross, Simon A. Levin and Joshua Weitz for important feedback and discussions. We would also like to thank Stefano Allesina and additional anonymous reviewers for their constructive input. E.B. acknowledges the support of the Technion Rothschild Scholars Program for Excellence. E.D.K. acknowledges government support by the Department of Defense, Office of Naval Research, National Defense Science and Engineering Graduate (NDSEG) Fellowship, 32 CFR 168a. R.K. acknowledges the support of European Research Council Seventh Framework Programme ERC Grant 281891, National Institutes of Health Grant R01GM081617, and Israeli Centers of Research Excellence I-CORE Program ISF Grant No. 152/11.

## Author contributions

E.B., E.D.K. and R.K. designed the study; E.B. did the simulations and analytical analysis; E.B., E.D.K. and R.K. analysed the data and wrote the paper.

## Additional information

**Competing financial interests:** The authors declare no competing financial interests.

