## [Peer Review file · Nature Communications]

Reviewers' Comments:

Reviewer #1 (Remarks to the Author)

This paper presents simulation results of multi-species models that incorporate terms to describe indirect effects where species modify direct interactions between pairs of species. The authors find that when these types of interactions are included, increasing complexity can promote species coexistence, in contrast to classic results obtained by May. Although there have been a few prior studies probing this issue, there remains much to do to understand the role that interaction modification play in ecosystems, and this research extends the generality of previously-noted effects on ecological complexity. I was provided the previous through reviews of the ms. and feel that the authors did a good job of addressing the issues they raised. There were a couple of points that I felt could still use some further clarification.

First, I thought there was ambiguity in the use of stability concepts and terms. May's original analysis focused on local point stability at equilibrium, whereas the analysis of the authors appears to focus on species persistence (feasibility). While the two may show similar behavior under linear interactions, it is known that this correspondence is not necessarily present for non-linear interactions, which of course include the higher-order/interaction modification cases of interest here. Hence, I was uncertain whether it is really appropriate to contrast the results with May. I guess the fact that the authors present simulations that attain similar results to May's while using (I think) their feasibility criteria would be an argument that comparisons to May are reasonable.

Second, I was uncertain, particularly after reading the responses to the reviewers, whether the authors are dealing with the underlying dynamic equations describing species interactions, or terms in the Jacobian matrices describing partial derivatives near equilibrium, which is what May was analyzing. As noted by one of the prior reviewers, mechanistic interpretation of the terms in a Jacobian matrix as higher-order is tricky, as off-diagonal terms in such a matrix from a system without higher-order effects may include the interaction coefficients of other species that influence the equilibrium values of the directly-interacting species described by these terms. It is also not clear to me how a simulation based on Jacobian matrices would be carried out. Based on the responses to reviewers and supplemental information, I was confused about what was being modeled. The authors should be careful to clearly distinguish when one or the other is being used. A small suggestion would also be to describe Equation (assuming I understand it correctly) as "per-capita growth rates" (line 59).

Third, the manuscript would be greatly strengthened if the authors could provide some insight into the mechanisms underpinning why their results were obtained. In particular, the pioneering study on this topic (Wilson 1992 Ecology) required spatial structure/sensitivity to initial conditions to generate the strongest positive diversity effects, yet this does not appear to be required in the results presented here. Why might this difference be obtained?

Reviewer #2 (Remarks to the Author)

A. Summary of the key results

This is a nice paper that I have enjoyed reading. In a simple and straightforward way, the authors show that the occurrence of high-order random interaction within communities set a lower bound to species diversity by making communities sensitive to species removal.

B. Originality and interest

While this work might seem close to previous results (de Oliveira & Fontanari, 2000) the analysis of the effect of high-order interactions on species diversity makes it a nice and novel contribution to the diversity-stability debate.

C. Data and methodology & D. Appropriate use of statistics and treatment of uncertainties

This paper already went through a process of review and the authors have carefully addressed each comments made by the three previous referees. The methodology is sound and can be replicated.

E. Conclusions

The conclusions of this paper are clearly drawn from the results of the simulations and made robust and reliable by the use of the additional simulations presented in the extended data.

F. Suggested improvements

It is my opinion that this paper do not need any further improvements.

G. References

The link between this work and the existing literature on this topic is clearly stated and appropriate credit is given to previous work.

H. Clarity and context

I found this manuscript straightforward to follow. In the past few years, numerous studies have helped advance the understanding of the relationship between the stability of ecological communities and their diversity. This study falls within this context and bring a new perspective to understanding how high-order interactions might influence this relationship.

Referees' comments:

Reviewer #1 (Remarks to the Author):

This paper presents simulation results of multi-species models that incorporate terms to describe indirect effects where species modify direct interactions between pairs of species. The authors find that when these types of interactions are included, increasing complexity can promote species coexistence, in contrast to classic results obtained by May. Although there have been a few prior studies probing this issue, there remains much to do to understand the role that interaction modification play in ecosystems, and this research extends the generality of previously-noted effects on ecological complexity. I was provided the previous through reviews of the ms. and feel that the authors did a good job of addressing the issues they raised. There were a couple of points that I felt could still use some further clarification.

First, I thought there was ambiguity in the use of stability concepts and terms. May's original analysis focused on local point stability at equilibrium, whereas the analysis of the authors appears to focus on species persistence (feasibility). While the two may show similar behavior under linear interactions, it is known that this correspondence is not necessarily present for non-linear interactions, which of course include the higher-order/interaction modification cases of interest here. Hence, I was uncertain whether it is really appropriate to contrast the results with May. I guess the fact that the authors present simulations that attain similar results to May's while using (I think) their

feasibility criteria would be an argument that comparisons to May are reasonable.

We thank the reviewer for appreciating the contribution of our work to the current knowledge regarding the role of interaction modifications in ecosystems. Indeed, May's analysis deals with local point stability at equilibrium, and it is not obvious that this local criterion sets a cap on biodiversity when an actual model is evolved in time. We now clarify that this cap on diversity was indeed observed in various models with pairwise interactions (first paragraph of introduction). This new comment will also allow readers to better understand what we mean by "May's analysis and subsequent results" in pg. 3. We thank the reviewer for allowing us to compare our results with previous findings in a more clear and accurate manner.

Second, I was uncertain, particularly after reading the responses to the reviewers, whether the authors are dealing with the underlying dynamic equations describing species interactions, or terms in the Jacobian matrices describing partial derivatives near equilibrium, which is what May was analyzing. As noted by one of the prior reviewers, mechanistic interpretation of the terms in a Jacobian matrix as higher-order is tricky, as off-diagonal terms in such a matrix from a system without higher-order effects may include the interaction coefficients of other species that influence the equilibrium values of the directly-interacting species described by these terms. It is also not clear to me how a simulation based on Jacobian matrices would be carried out. Based on the responses to reviewers and supplemental information, I was confused about what was being modeled. The authors should be careful to clearly distinguish when one or the other is being used.

We thank the reviewer for pointing out this confusing issue. We clarify that our results deal with the coefficients of the underlying dynamical equations and our theoretical analysis in the Methods section extends that of May's, using the Jacobian matrix and deriving the relation between the two. Following the reviewer comment and to avoid confusion, we have refrained from mentioning the comparison with May's result when introducing the model (p. 3 after eq. 1), and only make the comparison later on. We now also mention explicitly when we refer to the Jacobian (pg. 4 before eq. 2, also referred to as "the effective pairwise interactions considered in May's analysis" for the less technical readership).

A small suggestion would also be to describe Equation (assuming I understand it correctly) as "per-capita growth rates" (line 59).

Corrected.

Third, the manuscript would be greatly strengthened if the authors could provide some insight into the mechanisms underpinning why their results were obtained. In particular, the pioneering study on this topic (Wilson 1992 Ecology) required spatial structure/sensitivity to initial conditions to generate the strongest positive diversity effects, yet this does not appear to be required in the results presented here. Why might this difference be obtained?

Wilson works has not explicitly considered the scale of stability with number of species (He works with fixed number of 10 species). His result shows that in such conditions, 3-way interactions allow more diversity than comparable pairwise interactions in spatial environments due to the increased sensitivity to initial conditions.

We have now added a simple explanation to our results based on the expected variance of sum of random variables - see p. 4 and new Methods subsection in pages 9-10.

Reviewer #2 (Remarks to the Author):

A. Summary of the key results

This is a nice paper that I have enjoyed reading. In a simple and straightforward way, the authors show that the occurrence of high-order random interaction within communities set a lower bound to species diversity by making communities sensitive to species removal.

B. Originality and interest

While this work might seem close to previous results (de Oliveira & Fontanari, 2000) the analysis of the effect of high-order interactions on species diversity makes it a nice and novel contribution to the diversity-stability debate.

C. Data and methodology & D. Appropriate use of statistics and treatment of uncertainties

This paper already went through a process of review and the authors have carefully addressed each comments made by the three previous referees. The methodology is sound and can be replicated.

E. Conclusions

The conclusions of this paper are clearly drawn from the results of the simulations and made robust and reliable by the use of the additional simulations presented in the extended data.

F. Suggested improvements

It is my opinion that this paper do not need any further improvements.

G. References

The link between this work and the existing literature on this topic is clearly stated and appropriate credit is given to previous work.

H. Clarity and context

I found this manuscript straightforward to follow. In the past few years, numerous studies have helped advance the understanding of the relationship between the stability of ecological communities and their diversity. This study falls within this context and bring a new perspective to understanding how high-order interactions might influence this relationship.

We are delighted that the reviewer enjoyed reading our manuscript. We wish to thank the reviewer for appreciating our work and are grateful for their very kind words.